# PyraMotion: Attentional Pyramid-Structured Motion Integration for Co-Speech Gesture Synthesis

**Zhizhuo Yin**[1], **Yuk Hang Tsui**[2], **Pan Hui**[1,2]

[1]Hong Kong University of Science and Technology (Guangzhou) Guangzhou, Guangdong, China
[2]Hong Kong University of Science and Technology, Hong Kong SAR
`zyin190@connect.hkust-gz.edu.cn, yhtsui@connect.ust.hk, panhui@hkust-gz.edu.cn`

## Abstract

Generating full-body human gestures encompassing face, body, hands, and global movements from audio is crucial yet challenging for virtual avatar creation. Existing systems tokenize gestures with fixed frame-count for each token, predicting tokens of single scale from the input audio. However, expressive human gestures consist of varied patterns with different frame lengths, and different body parts exhibit motion patterns of varying durations. Existing systems fail to capture motion patterns across body parts and temporal scales due to the fixed frame-count setting of their gesture tokens. Inspired by the success of the feature pyramid technique in the multi-scale visual information extraction, we propose a novel framework named PyraMotion and an adaptive multi-scale feature capturing model called Attentive Pyramidal VQ-VAE (APVQ-VAE). Objective and subjective experiments demonstrate that the PyraMotion outperforms state-of-the-art methods in terms of generating natural and expressive full-body human gestures. Extensive ablation experiments highlight that the self-adaptiveness integration through attention maps contributes to performance.

## 1 Introduction

Among all human communication, the movements of all body gestures serve as an important approach to conveying thoughts. Such non-verbal signals provide more information than voice and context, enhancing the expressiveness and vividness of the speech, thus allowing listeners to gain a more comprehensive understanding of the intentions, emotions, and motivations of the speaker [32, 28]. Existing studies [35, 11] suggest that expressive gestures make avatars more intimate and trustworthy. In the metaverse [17], the naturalness of the avatar's full-body gestures, such as face micro-expressions, intricate body gestures, and movement trajectory, could impact the sense of realism, presence, and satisfaction. Therefore, generating natural full-body gestures is a valuable and challenging task that serves as a critical component for creating realistic digital humans [29, 42].

The most recent co-speech gesture generation work [22, 1, 39] focuses on utilizing VQ-VAE [34], a method that encodes continuous motion to a discrete latent space while preserving the original motion information. However, in these VQ-VAE-based methods, one shared approach is that each motion token exclusively represents a static human pose in a single scale with fixed frame count. Since natural gestures convey semantic information through a series of motion patterns with varying durations, shown in Figure 1, this assumption severely hampers the model's ability to learn natural gestures that express different intricate semantic information. This assumption also constrains the ability of the token predictor model to capture expressive patterns of audio and transcript representations with inconsistent durations and generate accurate corresponding motion tokens.

Inspired by a widespread pyramidal multiscale design exploited in computer vision [31, 20, 26]. We propose the PyraMotion framework to address the above problem by adaptively capturing motion

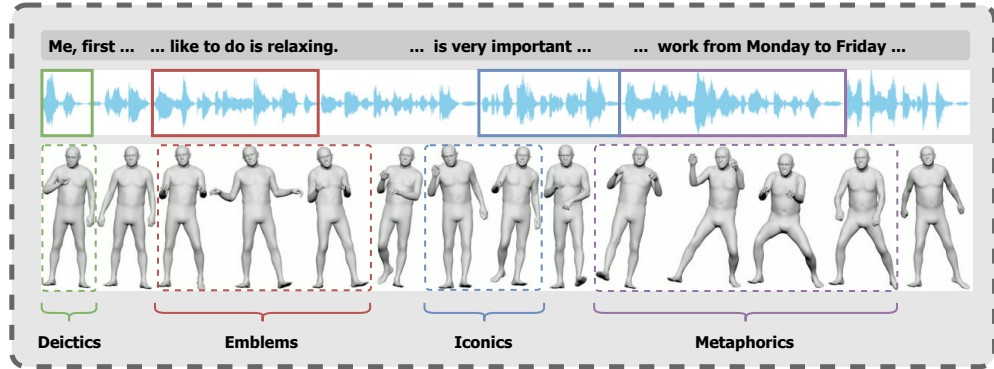

Figure 1: Illustration of gesture sequences with expressive motion patterns in different durations.

patterns from coarse to fine granularities. Experiments validate that PyraMotion outperforms existing state-of-the-art studies on reconstructing smooth body gestures from audio and textual information.

One of the key components of PyraMotion, the Attentional Pyramidal VQ-VAE (APVQ-VAE) model encodes the motion patterns of different body parts at different scales into a shared discrete latent space as several series of motion tokens and reconstructs a motion sequence by attentively fusing these motion tokens. Experiments also demonstrate that APVQ-VAE outperforms the traditional VQ-VAE in gesture reconstruction, beat alignment, and diversity. Additionally, the attention maps reveal that different body parts exhibit distinct attention patterns to tokens across varying time durations, highlighting the necessity to introduce a pyramidal motion representation.

To summarize, the main contributions are as follows:

- This work introduces a novel framework called the PyraMotion, which generates dynamic and natural full-body gestures using multi-modal information. To the best of our knowledge, this paper is the first to propose encoding motion patterns with varying temporal scale into a shared discrete latent space as motion pattern tokens.
- This work proposes the Attentional Pyramidal VQ-VAE (APVQ-VAE), a novel model designed to tokenize motion patterns across multiple temporal scales using a shared codebook. Experiments validate the superiority of APVQ-VAE compared with vanilla VQ-VAE in capturing motion patterns at different temporal scales and effectively accommodating the diverse motion information needs of various body parts during generation.
- Extensive experiments show that **Pyramotion** outperforms state-of-the-art methods in generating audio-driven gestures, as confirmed by both objective evaluations and subjective human studies.

## 2   Related Works

### 2.1   Human 3D Motion Generation Approaches

Early attempts at generating human gestures involved using rule-based algorithms [4, 15] to select appropriate gestures from a database based on input and blend motion clips with smooth algorithms. However, due to limited data and varying speech details, such a workflow suffers from inconsistency between the audio and motion, and the unnatural transition among different motion clips.

Deep generative models, such as Variational Autoencoders (VAEs) [16], Generative Adversarial Networks (GANs) [13], and transformers [36, 9], have shown promise in addressing the limitations of rule-based methods by capturing complex data correlations in a shared vector space. However, direct generation from a continuous latent space is sensitive to input noise, presenting challenges in practical applications. VAE-based methods also face the problem of "posterior collapse," restricting their ability to generate diverse gestures based on audio cues. These pose challenges for practical implementation in real-world scenarios.

Recent work [39, 30, 1, 22] has utilized VQ-VAE [34] to project motion patterns to a discrete latent codebook, transforming the motion generation problem into an autoregressive token prediction task conditioned on audio. These methods utilize VQ-VAE to capture motion patterns in single scale with fixed frame count and project them into a discrete latent space, represented by motion tokens.

Then, they utilize a token predictor network to transform the audio and text representations into the motion token series and reconstruct human motion of each frame by applying the decoder of VQ-VAE. While the VQ-VAE enhances model robustness and generation quality, the assumption that each token represents poses of gestures in a single scale limits the generation of expressive motions varying in frame counts.

To address the gap, this paper introduces a variation of VQ-VAE called Attentional Pyramidal VQ-VAE (APVQ-VAE) to tokenize dynamic motions in different temporal scales, enabling the generation of more detailed and natural gesture sequences.

### 2.2 Audio-driven Full-body Gesture Generation

Given the complexity and variability of human motions, recent research has focused on generating gestures for specific parts of the body, such as the face [10, 37], upper body (including arms, wrists, and hands) [41, 12], and lower body (including overall movements) [38] individually. While these generative models have shown promise in capturing unique motion patterns, they lack the versatility to generate movements across all body parts due to the diverse range of motion patterns among different body parts. There is a need for a unified framework for motion modeling.

Recent studies [22, 25, 6] approached generating full-body gestures by generating each part, the face, upper body, hands, lower body, and global movements first and combining these components to form full-body gestures.

However, existing methods overlook the need for different temporal information in modeling and generating motions, which lack the ability to capture motion patterns in different temporal scales and extract semantic correlations between different body parts.

In this work, we utilize the Attentional Pyramidal Token predictor to extract the coarse-to-fine-grained pattern information within the audio and transcript embeddings. It can accurately predict the motion token series by exploiting both body part information and semantic correlations between different body parts through a multi-scale temporal cross-attentive network.

## 3 Methodology

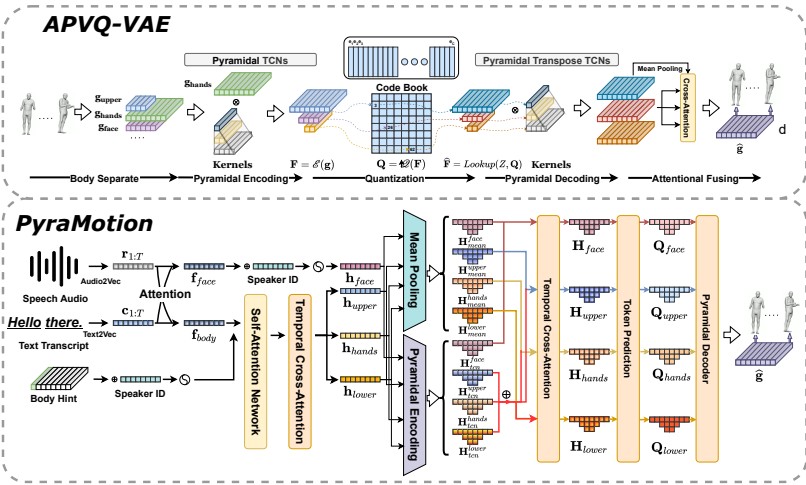

Figure 2: The Overall Workflow of PyraMotion. **Stage 1:** APVQ-VAE learns the discrete latent representations of motions, denoted as tokens, and reconstructs the motion from the pyradical token series via decoder. **Stage 2:** The PyraMotion framework is trained to predict the pyradical token series of motion from audio and reconstruct the motion via decoder in APVQ-VAE.

This section presents the development of the PyraMotion framework, as depicted in Figure 2. Initially, we formally define the problem of audio-driven gesture generation. Subsequently, we outline the workflow of the PyraMotion framework and delve into the structures of Attentional Pyramidal VQ-VAE and Pyramidal Token Predictor.

## 3.1 Problem Formulation

For a given audio $\mathbf{a} \in \mathbb{R}^{L \cdot sf}$, $sf = sr_{audio}/fps_{gestures}$, where $L$ represents the expected total frame counts of generated gesture sequences, $sf$ represents the sample count per frame, $sr_{audio}$ denotes the sample rate of the given audio, and $fps_{gestures}$ denotes the frames per second of the generate gesture sequences. The task is to generate the holistic 3D gesture of the human body $\mathbf{g} \in \mathbb{R}^{L \times (55 \times 6 + 100 + 4 + 3)}$ from the given audio. The dimension of $\mathbf{g}$ is formed by $L$ multiplied by the sum of 55 joints in Rot6D, 100 dimensions of facial expressions in FLAME parameters, 4 foot contact labels, and 3 global translation parameters.

## 3.2 PyraMotion

The overall workflow of PyraMotion can be divided into two stages: 1. Training APVQ-VAE to learn discrete latent representations of motions for encoding and reconstruction. 2. Training the Pyramidal Token Predictor to predict the discrete latent representations from audio.

In the first stage, motion representations are tokenized at various scales using multiple Temporal Convolution Networks (TCNs). These tokens are mapped to a shared codebook for semantic alignment, then reconstructed to original motions by applying Transpose TCN and aggregation.

In the second stage, the goal is to predict token series from input audio and text transcripts to reconstruct corresponding motion sequences. To account for the diverse motion patterns across different body parts, the audio and text representations are transformed into four separate latent embeddings for reconstructing the face, upper body, hands, and lower body. The four latent embeddings are further processed and quantized into token series of varying temporal scale, then decoded by the attentional pyramidal decoder in the trained APVQ-VAE to reconstruct the motion sequences.

## 3.3 Attentional Pyramidal VQ-VAE (APVQ-VAE)

Normal VQ-VAE consists of Encoder $\mathcal{E}$, Quantizer $\mathcal{Q}$, and Decoder $\mathcal{D}$. The Encoder projects the raw gesture sequences ($\mathbf{g}$) into the latent representation space as encoded embeddings $\mathbf{f}$ in single scale with fixed frame count, and then the Quantizer classifies the embeddings into discrete tokens $\mathbf{q}$ according to the codebook $Z$ based on the embedding distance in the latent space. After that, the decoder reconstructs the motion sequences $\hat{\mathbf{g}}$ from the quantized embedding $\hat{\mathbf{f}}$, which is generated by looking up the codebook with quantized token series.

$$\mathbf{f} = \mathcal{E}(\mathbf{g}), \mathbf{q} = \mathcal{Q}(\mathbf{f}), \hat{\mathbf{f}} = \text{lookup}(Z, \mathbf{q}), \hat{\mathbf{g}} = \mathcal{D}(\hat{\mathbf{f}})$$
$$\mathbf{f}, \hat{\mathbf{f}} \in \mathbb{R}^{T_{source} \times d}, \mathbf{q} \in \mathbb{R}^{T_{source}}, Z \in \mathbb{R}^{C \times d}, \mathbf{g}, \hat{\mathbf{g}} \in \mathbb{R}^{L \times (55 \times 6 + 100 + 4 + 3)} \tag{1}$$

where $T_{source}$, $d$, and $C$ are the total frame count of the input motion sequence, the dimensions of embeddings, and the size of the codebook, respectively.

In APVQ-VAE, we utilize $n$ TCN networks representing $n$ levels of temporal scale to generate the encoded embedding $\mathbf{F}$, which consists of $n$ embedding series with various lengths.

$$\mathbf{F} = [\mathbf{f}_1, ..., \mathbf{f}_n] = \mathcal{E}(\mathbf{g}) = [\text{TCN}_1(\mathbf{g}), ..., \text{TCN}_n(\mathbf{g})] \tag{2}$$

where $\mathbf{f}_i \in \mathbb{R}^{(T_{source}/2^i) \times d}$ denotes the embedding series with $i$ from 1 to $n$.

In the quantization stage, the encoded embeddings are first quantized to token series $\mathbf{Q}$, then formed the quantized embedding series $\hat{\mathbf{F}}$ by looking up the codebook. All embedding series share the same codebook $Z$ to ensure consistency in the embedding semantics.

$$\mathbf{Q} = [\mathbf{q}_1, ..., \mathbf{q}_n] = \mathcal{Q}(\mathbf{F}), \hat{\mathbf{F}} = \text{lookup}(Z, \mathbf{Q}) \tag{3}$$

where $\mathbf{q}_i \in \mathbb{R}^{T_{source}/2^i}$ denotes the token series.

In the decoding stage, instead of using TCN as previous VQ-VAE-based methods [22, 1], we utilize the Transpose Temporal Convolution Network (TransTCN) to decode the quantized embeddings into gesture sequences using the same scale of reception field in the encoder. Thus, multiple gesture sequences are reconstructed from quantized embeddings in different temporal scales. To further enhance the reconstruction, as shown in Eq. 4, a residual connection is introduced by combining the direct reconstruction from the mean of the stacked gesture sequences with the attention-based residue,

enabling the model to capture both coarse and fine-grained temporal dynamics effectively. Further details on this design and its effectiveness are discussed in section 4.5.

$$\hat{\mathbf{g}} = \mathcal{D}(\hat{\mathbf{F}}) = \text{Attention}(\text{Mean}(\hat{\mathbf{F}}_{stack}), \hat{\mathbf{F}}_{stack}, \hat{\mathbf{F}}_{stack}) + \text{Mean}(\hat{\mathbf{F}}_{stack})$$

$$\hat{\mathbf{F}}_{stack} = [\text{TransTCN}_1(\hat{\mathbf{f}}_1), ..., \text{TransTCN}_n(\hat{\mathbf{f}}_n)] \tag{4}$$

Notably, for each convolution operation in TCNs and TranTCNs, the output embeddings are normalized by dividing the kernel size shown in Eq. 5 to mitigate the scale differences stemming from varying kernel sizes, thereby enhancing the stability of the training process.

$$\mathbf{r} = Conv(\mathbf{f}, ks, s, p)/ks \tag{5}$$

where $Conv()$ denotes convolutional operation, $ks$ denotes kernel size, $s$ denotes stride, and $p$ denotes padding. The loss function of APVQ-VAE is as follows:

$$\mathcal{L}_{\text{VQ-VAE}} = \mathcal{L}_{rec}(\mathbf{g}, \hat{\mathbf{g}}) + \mathcal{L}_{vel}(\mathbf{g}', \hat{\mathbf{g}}')$$

$$+ \mathcal{L}_{acc}(\mathbf{g}'', \hat{\mathbf{g}}'') + ||sg[\mathbf{F}] - \hat{\mathbf{F}}||_2^2 + ||\mathbf{F} - sg[\hat{\mathbf{F}}]||_2^2 \tag{6}$$

where $\mathcal{L}_{rec}$ is Geodesic [33] loss, and $\mathcal{L}_{vel}$, $\mathcal{L}_{acc}$ are L1 losses. sg is the stop gradient operation and the weight of commitment [34] loss is set to 1.

## 3.4 Pyramidal Token Predictor

In the Pyramidal Token Predictor, we need to predict the corresponding pyramidal tokens series $\hat{\mathbf{Q}} = [\hat{\mathbf{q}}_1, ..., \hat{\mathbf{q}}_n]$ from audio features and text contents.

**Audio and Context Feature Fusion**    For the input audio features $\mathbf{s}$, following previous work [2], we employ onset $\mathbf{o}$, and amplitude $\mathbf{a}$ and combine them as rhythmic audio features $\mathbf{r} \in \mathbb{R}^{T \times d}$. For the input transcript text, we transform it into content features $\mathbf{c} \in \mathbb{R}^{T \times d}$ using the pre-trained model [3]. Then we utilize the attention mechanism to fuse the audio features and context features,

$$\alpha = \text{Softmax}(\text{MLP}(\mathbf{r}_{1:T}, \mathbf{c}_{1:T}))$$

$$\mathbf{f}_{1:T} = \alpha \times \mathbf{r}_{1:T} + (1 - \alpha) \times \mathbf{c}_{1:T} \tag{7}$$

where $T$ denotes the total frame counts of the audio after sampling and the expected motion sequences. $\alpha \in \mathbb{R}^{T \times d}$ is the element-wise attention coefficient.

**Latent Generation and Token Prediction**    According to the previous studies [22], in the holistic gesture generation task, there is only a weak correlation between the distribution of facial expression and body motion. Therefore, we propose to exploit two independent feature extraction and token prediction workflows for facial expression and body motion separately, based on independent audio-text representations $\mathbf{f}^{\mathbf{face}}$ and $\mathbf{f}^{\mathbf{body}}$. For the latent representation prediction, we first combine the audio-text embedding with the learned speaker embedding and project to the dimension of the hidden vector:

$$\mathbf{h}^{face} = \text{MLP}(\mathbf{f}^{face} \oplus \mathbf{p}_f)$$

$$\mathbf{h}^{hints} = \text{SAN}(\bar{\mathbf{g}} + \mathbf{p}_t) \tag{8}$$

$$\mathbf{h}^{body} = \mathbf{h}^{hints} + \text{TCAT}(\mathbf{h}^{hints} \oplus \mathbf{p}_t, \mathbf{f}^{body})$$

$\mathbf{h}^{parts} \in \mathbb{R}^{T \times h}$ denotes the hidden representation of part motion, where $parts$ can be $face$ or $body$. $\mathbf{f}^{parts} \in \mathbb{R}^{T \times d}$ denotes the fused audio-text embedding, and $\mathbf{p}_f \in \mathbb{R}^d$ denotes the learned speaker embedding. Body hints $\mathbf{h}^{hints} \in \mathbb{R}^{T \times h}$ are encoded from the masked gesture sequence $\bar{g}$, where the first 8 frames are ground truth frames. SAN denotes a self-attentive network.

To better capture the various motion patterns of different parts of the body, the hidden representation of the body is further projected into three different latent spaces, including the upper body, lower body, and hands. Then, we construct two types of pyramidal representations to generate pyramidal latent. The first type averages the consecutive representation and the second type uses the isomorphic TCN network in the APVQ-VAE encoder. The TCN network can extract similar information as the APVQ-VAE encoder and generate pyramidal motion latent.

$$\mathbf{h}^{upper,lower,hands} = \text{MLP}(\mathbf{h}^{body})$$

$$\mathbf{H}^{parts}_{mean} = [\mathbf{h}^{parts}_{mean_0}, ..., \mathbf{h}^{parts}_{mean_{n-1}}] = [\sigma(\mathbf{h}^{parts}_{[0,..,2^i-1]}), ..., \sigma(\mathbf{h}^{parts}_{[n-2^i,..,n-1]})] \tag{9}$$

$$\mathbf{H}^{parts}_{tcn} = [\mathbf{h}^{parts}_{tcn_0}, ..., \mathbf{h}^{parts}_{tcn_{n-1}}] = [\text{TCN}_0(\mathbf{h}^{parts}), ..., \text{TCN}_{n-1}(\mathbf{h}^{parts})]$$

where $parts$ can be $face$, $upper$, $lower$, and $hands$. $\sigma$ denotes the average operation and $\mathbf{h}^{parts}_{mean\,i}, \mathbf{h}^{parts}_{tcn\,i} \in \mathbb{R}^{T/2^i \times h}$ denotes the $i^{th}$ embedding series with $2^i$ frames.

After constructing the embedding series with different scale, a temporal cross-attention Transformer decoder TCAT is applied to capture the correlations between the above two representations. Then a 1-layer MLP is used to project the dimensions into the reconstruction latent $\hat{\mathbf{H}}^{parts}_{rec}$.

$$\hat{\mathbf{H}}^{parts}_{rec} = \mathrm{MLP}(\mathrm{TCAT}(\mathbf{H}^{parts}_{mean}, \mathbf{H}^{parts}_{tcn})) \tag{10}$$

We optimize the learned reconstruction latent by applying the MSE loss:

$$\mathcal{L}^{parts}_{rec} = \mathrm{MSELoss}(\hat{\mathbf{H}}^{parts}_{rec}, \hat{\mathbf{F}}^{parts}) \tag{11}$$

where $\hat{\mathbf{F}}^{parts}$ is the learned latent representation of APVQ-VAE of corresponding body parts from the ground truth. After learning the mutual information between the two representations, we further learned the correlation among body parts. We sum the TCN latent series of all body parts together to form a full-body pyramidal latent and conduct cross-attention operations with the mean latent:

$$\begin{aligned} \mathbf{H}^{full} &= \mathbf{H}^{upper}_{tcn} + \mathbf{H}^{hands}_{tcn} + \mathbf{H}^{lower}_{tcn} \\ \tilde{\mathbf{H}}^{parts} &= \mathrm{TCAT}(\mathbf{H}^{parts}_{mean}, \mathbf{H}^{full}) = [\tilde{\mathbf{h}}^{parts}_0, ..., \tilde{\mathbf{h}}^{parts}_{n-1}] \end{aligned} \tag{12}$$

where $\tilde{\mathbf{H}}^{parts}$ denotes the latent embedding of each part of the body in different temporal scale and $parts$ can be $upper$, $lower$, and $hands$ according to previous body part separation. $\mathbf{h}^{parts}_i \in \mathbb{R}^{T/2^i \times h}$ represents the latent embedding of body parts in different scales.

After generating the latent representations of each body part, we quantize the latent representations by predicting the corresponding tokens iteratively from coarse to fine granularity. Such quantization operation projects the learned continuous latent to discrete latent space represented by tokens for better reconstruction performance.

$$\begin{aligned} \hat{\mathbf{q}}^{parts}_i &= \mathrm{MLP}(\tilde{\mathbf{h}}^{parts}_i + \hat{\mathbf{q}}^{parts}_{i+1}), i \in [0, n-2] \\ \hat{\mathbf{Q}}^{parts} &= [\hat{\mathbf{q}}^{parts}_0, ..., \hat{\mathbf{q}}^{parts}_{n-1}] \end{aligned} \tag{13}$$

where $\hat{\mathbf{q}}^{parts}_i \in \mathbb{R}^{T/2^i \times C}$ represents the predicted possibility distribution of motion tokens for $i^{th}$ granularity, $C$ denotes the codebook size. We use cross-entropy loss to optimize the learned token index distribution, where $\mathbf{Q}^{parts}$ is the learned token of body motion in APVQ-VAE.

$$\mathcal{L}^{parts}_{cls} = \mathrm{CrossEntropy}(\hat{\mathbf{Q}}^{parts}, \mathbf{Q}^{parts}) \tag{14}$$

## 4 Experiments

### 4.1 Datasets and Training

We evaluate the ability of our method to generate holistic 3D gestures from speech on a diverse and expressive dataset BEAT2[1] [23] collected from mocap equipment. This public dataset contains 76 hours of high-quality, multi-modal data captured from 30 speakers talking with eight different emotions. Following the settings of existing work [22, 25], we conduct the experiments on the BEAT2-Standard Speaker2 with an 85%/7.5%/7.5% train/val/test split.

### 4.2 Evaluation Metrics

For the body motion generation, we adopt **FGD** [41] to measure the similarity between the generated gesture and the real gestures. To evaluate the **Diversity** [18] of generated gestures, we calculate the L1 distance between different gesture clips. In terms of audio-motion synchronization, we utilize the **Beat Align** [19] measurement. For the facial expression generation, we use **Mean Square Error (MSE)** [37] to measure the vertex position distance and **L1 Vertex Difference (LVD)** [37] to measure the L1 distance between the generated facial expression and the ground truth facial expression. The results are reported as the mean value and the standard deviation computed from 5 times of independent runs. The significance is also reported with the p-value.

---

[1]https://github.com/PantoMatrix/PantoMatrix

## 4.3 Comparison Methods

We compare our PyraMotion[2] with the following classic and state-of-the-art methods of talking head generation and body gesture generation: S2G [12], Trimodal [41], HA2G [24], DisCo [21], CaMN [23], Diffusestylegesture [38], HoloGest [7],RAG-Gesture [27], AMUSE [8], MambaTalk [43], Habibie et al [14],TalkSHOW [40], DiffSHEG [6], EMAGE [22], and ProbTalk [25], SynTalker [5]. For Habibie et al. [14], TalkSHOW [40], we use their reported version, which is the upper body generation. Meanwhile, we cite the results of their full-body version reproduced by [22].

## 4.4 Overall Comparison

In this part, we compare the overall performances of M3G with classical and state-of-the-art audio-driven gesture generation methods. In Table 1, **Habibie et al**[‡] and **TalkSHOW**[‡] denotes the reported performance of reproduced full-body motion generation in [22], * denotes the results are directly adapted from their original paper due to the same experimental settings, thus no std values are reported. **AMUSE**[†] denotes the reported performance is by the reproduced evaluation code by ourselves.

| | FGD $^{\times 10^{-1}}\downarrow$ | BA $^{\times 10^{-1}}\rightarrow$ | Diversity$\rightarrow$ | MSE$^{\times 10^{-8}}\downarrow$ | LVD$^{\times 10^{-5}}\downarrow$ |
|---|---|---|---|---|---|
| **S2G (CVPR 2019)** | 27.87 | $4.827 \pm 0.138$ | $6.022 \pm 0.097$ | - | - |
| **Trimodal (TOG 2020)** | 12.13 | $5.762 \pm 0.063$ | $7.513 \pm 0.073$ | - | - |
| **HA2G (CVPR 2022)** | 12.32 | $6.779 \pm 0.021$ | $8.626 \pm 0.016$ | - | - |
| **DisCo (ACM MM 2022)** | 9.484 | $6.439 \pm 0.027$ | $9.912 \pm 0.022$ | - | - |
| **CaMN (ECCV 2022)** | 6.967 | $6.628 \pm 0.018$ | $11.18 \pm 0.089$ | - | - |
| **DiffStyleGesture (IJCAI 2023)** | 8.866 | $7.239 \pm 0.089$ | $11.13 \pm 0.077$ | - | - |
| **AMUSE**[†] **(CVPR 2024)** | 12.11 | $\mathbf{8.318 \pm 0.052}$ | $\mathbf{14.93 \pm 1.497}$ | - | - |
| **SynTalker*** **(MM 2024)** | 5.366 | 7.812 | 13.05 | - | - |
| **HoloGest*** **(3DV 2025)** | 5.341 | *7.957* | *14.15* | - | - |
| **RAG-Gesture*** **(CVPR 2025)** | 8.08 | 7.34 | 11.97 | - | - |
| **Habibie et al**[‡] **(IVA 2021)** | 9.040 | 7.716 | 8.213 | 8.614 | 8.043 |
| **TalkSHOW (CVPR 2023)** | 6.145 | $6.863 \pm 0.008$ | $13.12 \pm 0.156$ | $7.791 \pm 0.044$ | $7.771 \pm 0.052$ |
| **TalkSHOW**[‡] **(CVPR 2023)** | 6.209 | 6.947 | 13.47 | 7.791 | 7.771 |
| **DiffSHEG*** **(CVPR 2024)** | 8.986 | 7.142 | 11.91 | 7.665 | 8.673 |
| **EMAGE (CVPR 2024)** | 5.643 | $7.707 \pm 0.004$ | $12.92 \pm 0.198$ | $7.694 \pm 0.076$ | $7.593 \pm 0.062$ |
| **ProbTalk*** **(CVPR 2024)** | 5.040 | 7.711 | 13.27 | 8.617 | - |
| **MambaTalk*** **(MMM 2025)** | 5.366 | 7.812 | 13.05 | $\mathbf{6.289}$ | $\mathbf{6.897}$ |
| **PyraMotion** | $\mathbf{4.612}$ | $7.420 \pm 0.008$ | $13.42 \pm 0.020$ | *7.176 ± 0.028* | *7.270 ± 0.011* |
| *p-value* | - | $< 0.0005$ | $< 0.0001$ | $< 0.0001$ | $< 0.001$ |

Table 1: Overall Comparison of various methods. The best performance of each metric is in **boldface** fonts, and the second one is in *font*. The sign $\uparrow$ denotes that the larger the value, the better it is, while the sign $\downarrow$ is the reverse, $\rightarrow$ means the metrics measures some expressive aspects of the motion, while higher value is not necessarily correlated to the generation accuracy. The standard deviation is calculated across 10 epochs after reaching best performance.

As shown in Table 1, our proposed **PyraMotion** surpasses or achieves state-of-the-art methods across all metrics. Notably, it achieves significant improvements in reconstruction accuracy like **FGD**, **MSE**, and **LVD**, demonstrating that its generated full-body gestures align more closely with ground-truth motion than existing approaches. For non-deterministic metrics like **Beat Align** and **Diversity**, **PyraMotion** delivers comparable or suboptimal performance.

These results highlight that PyraMotion's use of Attentional Pyramidal VQ-VAE to adaptively capture multi-scale patterns across body parts enables it to model a broader range of motion patterns. This capability enhances the model's ability to generate higher-quality, more accurate motion sequences compared to other methods.

## 4.5 Ablation Analysis

**Key components** In this ablation experiment, we propose five variants of **PyraMotion**: **w/o Separate Body** uses unified latent embeddings for different body parts instead of separate latent

---

[2]The source code will be released to GitHub after acceptance.

representations. **w/o Text** generates motion sequences only based on the audio signal. **w/o TCN Encoder** discards the TCN network in equation (12) to disable the temporal information extracting process in the token predictor; **w/o Full-Body Latent** discards the fusing operation of full-body latent in equation (15) and generates each body part's latent embedding only based on their own parts' features; **w/o TransTCN** substitutes the TransTCN structure in equation (4) with the TCN structure in equation (2) following previous work [22, 1].

Table 2 reported that the **PyraMotion** significantly outperforms or performs similarly with all variants, demonstrating the contributions of these components. Moreover, the results indicate that replacing the **TransTCN** structure leads to a more significant decline in performance compared with the other two variants, demonstrating the indispensability of the token decoding process in the overall workflow. The absence of **Full-Body latent** mainly affects the body's reconstructing performance, which might be caused by the lack of mutual information among different body parts. The **w/o TCN Encoder** performs significantly worse than **PyraMotion** on facial reconstruction and body diversity, indicating the ability of the Pyramidal TCN Encoder to model diverse types of motion patterns and facial expressions from audio.

| | $\text{FGD}^{\times 10^{-1}}\downarrow$ | $\text{BC}^{\times 10^{-1}}\uparrow$ | Diversity$\uparrow$ | $\text{MSE}^{\times 10^{-8}}\downarrow$ | $\text{LVD}^{\times 10^{-5}}\downarrow$ |
|---|---|---|---|---|---|
| **w/o Separate Body** | $8.926 \pm 0.021$ | $6.843 \pm 0.008$ | $10.838 \pm 0.024$ | $9.243 \pm 0.013$ | $8.153 \pm 0.019$ |
| **w/o Text** | $6.677 \pm 0.023$ | $7.374 \pm 0.012$ | $12.893 \pm 0.043$ | $8.612 \pm 0.012$ | $7.852 \pm 0.011$ |
| **w/o TransTCN** | $6.178 \pm 0.035$ | $7.132 \pm 0.005$ | $12.869 \pm 0.057$ | $8.833 \pm 0.017$ | $9.511 \pm 0.015$ |
| **w/o Full-Body Latent** | $5.152 \pm 0.034$ | $7.232 \pm 0.002$ | $13.308 \pm 0.062$ | $7.440 \pm 0.013$ | $7.518 \pm 0.012$ |
| **w/o TCN Encoder** | $5.178 \pm 0.013$ | $\mathbf{7.436 \pm 0.004}$ | $12.439 \pm 0.117$ | $7.607 \pm 0.081$ | $7.589 \pm 0.039$ |
| **PyraMotion** | $\mathbf{4.612 \pm 0.014}$ | $7.420 \pm 0.008$ | $\mathbf{13.42 \pm 0.020}$ | $\mathbf{7.176 \pm 0.028}$ | $\mathbf{7.270 \pm 0.011}$ |
| *p-value* | $< 0.0005$ | $> 0.1$ | $< 0.0001$ | $< 0.0001$ | $< 0.0001$ |

Table 2: Ablation Experiments for Proposed Components.

**Reconstruction Performance Comparison of APVQ-VAE** This section evaluates the performance of APVQ-VAE in learning and reconstructing compared to the widely used VQ-VAE in existing methods. A variant of it called **Mean Pyramidal VQ-VAE (MPVQ-VAE)** is also introduced, which substitutes the attentional fusing operation on the pyramidal token series with an averaging operation.

Table 3 presents the joints' rotation Mean Square Error (JRMSE) for each body part compared to the ground truth sequences. The second part of the table shows the metrics used in Table 1 to assess the reconstruction performance based on the encoded tokens in APVQ-VAE.

| | $\text{Face}^{\times 10^{-3}}\downarrow$ | $\text{Upper}^{\times 10^{-2}}\downarrow$ | $\text{Hands}^{\times 10^{-2}}\downarrow$ | $\text{Lower}^{\times 10^{-2}}\downarrow$ | $\text{Global}^{\times 10^{-2}}\downarrow$ | $\text{FGD}^{\times 10^{-1}}\downarrow$ | $\text{BC}^{\times 10^{-1}}\uparrow$ | Diversity$\uparrow$ | $\text{MSE}^{\times 10^{-8}}\downarrow$ | $\text{LVD}^{\times 10^{-5}}\downarrow$ |
|---|---|---|---|---|---|---|---|---|---|---|
| **VQ-VAE** | $2.100 \pm 0.009$ | $5.209 \pm 0.028$ | $7.103 \pm 0.017$ | $3.335 \pm 0.023$ | $4.418 \pm 0.011$ | $3.302 \pm 0.036$ | $\mathbf{7.488 \pm 0.014}$ | $12.482 \pm 0.009$ | $0.524 \pm 0.002$ | $2.087 \pm 0.010$ |
| **MPVQ-VAE** | $1.368 \pm 0.022$ | $2.972 \pm 0.016$ | $4.818 \pm 0.024$ | $2.302 \pm 0.025$ | $\mathbf{4.103 \pm 0.020}$ | $1.497 \pm 0.017$ | $7.143 \pm 0.012$ | $12.654 \pm 0.008$ | $0.441 \pm 0.005$ | $2.011 \pm 0.012$ |
| **APVQ-VAE** | $\mathbf{1.044 \pm 0.022}$ | $\mathbf{2.955 \pm 0.018}$ | $\mathbf{4.662 \pm 0.011}$ | $\mathbf{2.209 \pm 0.025}$ | $5.129 \pm 0.031$ | $\mathbf{1.296 \pm 0.028}$ | $7.237 \pm 0.015$ | $\mathbf{12.864 \pm 0.008}$ | $\mathbf{0.279 \pm 0.005}$ | $\mathbf{1.525 \pm 0.019}$ |
| *p-value* | $< 0.0001$ | $< 0.0001$ | $< 0.0005$ | $< 0.0001$ | $< 0.0001$ | $< 0.0001$ | $< 0.0001$ | $< 0.0001$ | $< 0.0001$ | $< 0.0001$ |

Table 3: Experiments for Reconstruction Errors.

The experimental results reveal that both our APVQ-VAE and its mean-fusing counterpart MPVQ-VAE significantly outperform conventional VQ-VAE in motion pattern tokenization and reconstruction, confirming the superiority of modeling through pyramidal token series.

Notably, APVQ-VAE surpasses the mean-fusing version, MPVQ-VAE, underscoring the importance of its attentional fusing operation across multiple scales. To interpret this mechanism, we visualize attention maps from the best epoch (lowest JRMSE) in Figure 3, revealing distinct scale preferences across body parts. The face predominantly utilizes fine-grained features (kernel size 1), likely due to continuous mouth motions. In contrast, the lower body balances attention between kernel sizes 2 and 8, corresponding to coarse leg dynamics. Meanwhile, hands and upper body exhibit decreasing preference from fine to coarse scales with attention weights ranging from 0.2 to 0.3, reflecting their need for both expressive details and structural motion patterns. These findings demonstrate that the attentional fusion mechanism in APVQ-VAE adaptively addresses the differential reliance on multi-scale features across distinct body regions.

**Efficiency Analysis** In this section, we will provide the theoretical analysis towards the computational complexity of the proposed workflow compared with vanilla VQ-VAE.

Theoretically, the computational complexity of our APVQ-VAE is:

$$O\left(\sum_{p=1}^{N} (2^p d)\left(\frac{Nd}{2^p}\right)\right) = O(PNd^2)$$

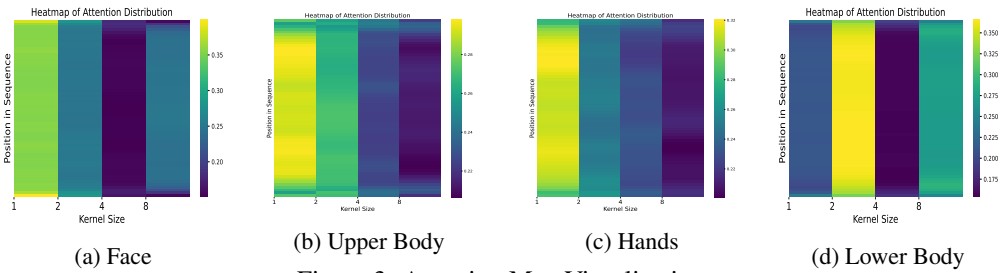

| (a) Face | (b) Upper Body | (c) Hands | (d) Lower Body |

Figure 3: Attention Map Visualization

where $d$ is the latent dimension, $N$ is the total number of frames, and $P$ is the number of pyramid layers. Thus, the time complexity of APVQ-VAE is a constant multiple of vanilla VQ-VAE.

For the PyraMotion model, before the pooling operation, the data processing steps are identical with single scale version. After the pooling operation described in equation Eq. 9, the time complexity in each step is in multi-scales, which follows the rule of a geometric series with regard to the increasing scale numbers. For the Temporal Cross-Attention (TCA) operation, the time complexity of its single-scale version is:

$$O(Nd^2 + N^2d)$$

where $N$ is the clip length and $d$ is the hidden dimension. The multi-scale Temporal Cross-Attention operation has complexity:

$$O\left(\sum_{p=0}^{P}\left(\frac{Nd^2}{2^p} + \left(\frac{N}{2^p}\right)^2 d\right)\right) \subseteq O\left(2Nd^2 + \frac{4N^2d}{3}\right) = O(2(Nd^2 + N^2d)) = O(Nd^2 + N^2d)$$

A similar conclusion will also be reached on the other multi-scale operations. Considering the total time complexity before the pooling operation, PyraMotion's time complexity has the same time-complexity level with less than $2\times$ of the single-scale version. To validate these theoretical conclusions, we measured the inference times for single-scale EMAGE and multi-scale PyraMotion on over 5 minutes of motion data. The experimental results are shown in Table 4.

Table 4: Inference Time Comparison

| Method | Inference Time |
|---|---|
| EMAGE | $22 \pm 1.143$ s |
| PyraMotion | $41 \pm 1.822$ s |

**Selection of Pyramid Layer Number** This section evaluates the influence of pyramid layer design on motion reconstruction performance. Adhering to the feature pyramid framework [20], which leverages exponentially increasing kernel sizes to reduce information overlap across layers, we configure successive layers with the scale sequence [1, 2, 4, 8, 16]. To preserve information density and mitigate noise introduced by the padding operation, the pyramid layer downsamples features by a factor of 2, corresponding to the dilation of their kernel sizes shown in Figure 2.

In this experiment, we search for the optimal number of layers to balance computational efficiency with hierarchical feature extraction. The results, summarized in Table 5, show that increasing the layer number initially boosts performance but eventually leads to a decline. More layers enable PyraMotion with a higher capability to represent complex motion patterns with varying durations. However, overcomplex pyramidal token series may cause overfitting and the difficulty in predicting tokens from audio, resulting in a comparative reconstruction performance with far worse generation performance from audio, such as the **5 Layers** variant.

| | $\textbf{Face}^{\times 10^{-3}}\downarrow$ | $\textbf{Upper}^{\times 10^{-2}}\downarrow$ | $\textbf{Hands}^{\times 10^{-2}}\downarrow$ | $\textbf{Lower}^{\times 10^{-2}}\downarrow$ | $\textbf{Global}^{\times 10^{-2}}\downarrow$ | $\textbf{FGD}^{\times 10^{-1}}\downarrow$ | $\textbf{BC}^{\times 10^{-1}}\uparrow$ | $\textbf{Diversity}\uparrow$ | $\textbf{MSE}^{\times 10^{-8}}\downarrow$ | $\textbf{LVD}^{\times 10^{-5}}\downarrow$ |
|---|---|---|---|---|---|---|---|---|---|---|
| **1 Layer** | $2.100 \pm 0.009$ | $5.209 \pm 0.028$ | $7.103 \pm 0.017$ | $3.335 \pm 0.023$ | $\textbf{4.418} \pm \textbf{0.011}$ | $3.302 \pm 0.036$ | $\textbf{7.488} \pm \textbf{0.014}$ | $12.482 \pm 0.009$ | $0.524 \pm 0.002$ | $2.087 \pm 0.017$ |
| **2 Layers** | $1.667 \pm 0.026$ | $3.262 \pm 0.022$ | $5.949 \pm 0.015$ | $2.546 \pm 0.027$ | $4.983 \pm 0.019$ | $6.768 \pm 0.026$ | $7.342 \pm 0.027$ | $12.979 \pm 0.005$ | $0.411 \pm 0.005$ | $1.858 \pm 0.010$ |
| **3 Layers** | $1.352 \pm 0.038$ | $2.993 \pm 0.025$ | $5.218 \pm 0.021$ | $2.484 \pm 0.035$ | $5.021 \pm 0.018$ | $5.627 \pm 0.017$ | $7.254 \pm 0.022$ | $\textbf{13.723} \pm \textbf{0.011}$ | $0.391 \pm 0.010$ | $1.616 \pm 0.016$ |
| **4 Layers** | $\textbf{1.044} \pm \textbf{0.022}$ | $\textbf{2.612} \pm \textbf{0.023}$ | $\textbf{4.662} \pm \textbf{0.011}$ | $\textbf{2.209} \pm \textbf{0.025}$ | $5.129 \pm 0.031$ | $\textbf{1.296} \pm \textbf{0.028}$ | $7.237 \pm 0.015$ | $12.864 \pm 0.008$ | $\textbf{0.279} \pm \textbf{0.005}$ | $\textbf{1.525} \pm \textbf{0.019}$ |
| **5 Layers** | $1.508 \pm 0.022$ | $2.955 \pm 0.018$ | $5.483 \pm 0.014$ | $2.339 \pm 0.029$ | $5.211 \pm 0.023$ | $6.463 \pm 0.017$ | $6.982 \pm 0.018$ | $12.263 \pm 0.008$ | $0.389 \pm 0.005$ | $1.658 \pm 0.012$ |
| *p-value* | $< 0.0001$ | $< 0.0001$ | $< 0.0005$ | $< 0.0001$ | $< 0.0001$ | $< 0.0001$ | $< 0.0001$ | $< 0.0001$ | $< 0.0001$ | $< 0.0001$ |

Table 5: Experiments for Pyramid Layer Number Selection

# 5 Perceptual Study

To evaluate the naturalness of our generated results from the subjective perception, we conducted A/B testing following [40] by comparing the generated motions by PyraMotion with other methods,

including **CaMN**, **EMAGE**, **APVQ-VAE Reconstruction**, and **Ground Truth**. For **CaMN**, since it does not generate the facial expression, we directly use the ground truth as its facial expression.

We randomly sampled 40 clips from the generated video and compared them pairwisely. Eighteen participants took part in this study. Specifically, participants are asked to answer A or B to the following questions: Which clip do you think is more natural?

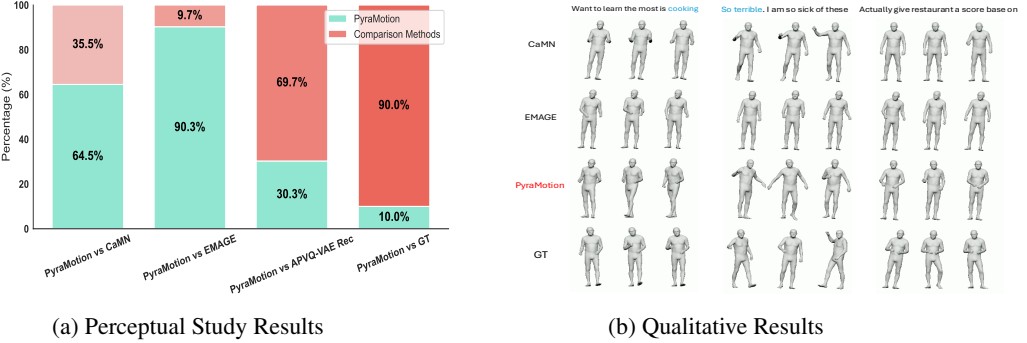

(a) Perceptual Study Results          (b) Qualitative Results

Figure 4: Perceptual Study

Figure 4a illustrates that participants favor **PyraMotion** over existing work **CaMN**, and **EMAGE**, indicating the superiority of **PyraMotion** in generating more natural motion from audio. Moreover, in a proportion of cases, **PyraMotion** outperforms the **APVQ-VAE Reconstruction**, which generates the motion from encoded tokens in APVQ-VAE, showing the closeness of generation quality between them. **GT** outperforms **PyraMotion** without a doubt, showcasing the space for improvements.

## 6 Limitation, Future Work

The limitations of the proposed method are as follows: 1) It cannot adaptively adjust the number of pyramid layers towards different datasets. The number of pyramid layers is a hyperparameter in this work, which might restrict its generalizability and usability towards real-world usage scenarios. 2) One observation based on the existing work, EMAGE, and our work is that overly focusing on integrating the correlation among different body parts could potentially cause a decrease due to the interference of different body parts' motion distributions. While fully independent modeling of each body part would cause unnatural behavior. Thus, a self-adaptive mechanism of adjusting the ratio between these two perspectives would be potentially beneficial for future studies.

## 7 Conclusion

This study introduces an audio-driven holistic gesture synthesis framework named **PyraMotion**, a method that can extract pyramidal features from audio and exploit these features to generate natural and realistic holistic 3D gestures. To achieve this, we propose the **APVQ-VAE**, which encodes the pyramidal feature of multi-scale motion patterns into a shared codebook and adaptively fuses these features for different body parts. We validate the effectiveness of **APVQ-VAE** by comparing its reconstruction performance with vanilla VQ-VAE. We also illustrate the attention map of pyramidal features fusion in **APVQ-VAE**. It supports the observation of varying preferences of motion scales across body parts and interprets the source of adaptivity in **APVQ-VAE**. Extensive experiments demonstrate the superiority of the **PyraMotion** framework in generating more natural and realistic holistic 3D gestures compared with current state-of-the-art methods.

## 8 Acknowledgement

This work is supported by both the Bureau of Science and Technology of Nansha District, Guangzhou, Grant No.2022ZD012, and HSBC Project L0562.

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

## A    Training Details

The whole training process is conducted on an Ubuntu Server with 1 GPU computing card with 32 GB VRAM and 256 GB memory. The average training time of APVQ-VAE of different body parts is around 24 hours, and the PyraMotion framework takes 48 hours to achieve best performance. For the software environment, the model is deployed using Python 3.9, PyTorch 2.4.1.

