# OpenReview forum: "PyraMotion: Attentional Pyramid-Structured Motion Integration for Co-Speech 3D Gesture Synthesis"
_NeurIPS.cc/2025/Conference — NeurIPS 2025 poster_

### Official Review · Reviewer_CqFS · 2025-06-21

**Clarity:** 3
**Significance:** 2
**Originality:** 2
**Rating:** 4
**Confidence:** 4

**Summary:**

PyraMotion: Attentional Pyramid-Structured Motion Integration for Co-Speech Gesture Synthesis
In this paper, the authors propose to use multi-scale part separated tokens to learn the embedding of co-speech motions (gesture specifically).

**Questions:**

1. How fast is the runtime inference time for the algorithm? Is it streamable?

2. The claim that “Existing systems tokenize gestures frame-wise” is incorrect or inaccurate.
In [1], [2], [3], [4], [5] and other existing work, tokens are usually generated with a 1D convolutionary network. It usually has a down-sampling rate of for example, 4, and each token has a much bigger perception field than 4.
And it is also worthwhile to point out that in [2] the authors also use separate tokenization for different body parts (upper body and lower body).
In my own experience there’s no benefit to use separate tokenizations, which is why I mentioned above that some ablation or comparisons between tokenization strategy is needed.

[1] Guo, Chuan, Yuxuan Mu, Muhammad Gohar Javed, Sen Wang, and Li Cheng. "Momask: Generative masked modeling of 3d human motions." In Proceedings of the IEEE/CVF Conference on Computer Vision and Pattern Recognition, pp. 1900-1910. 2024.

[2] Pinyoanuntapong, Ekkasit, Pu Wang, Minwoo Lee, and Chen Chen. "Mmm: Generative masked motion model." In Proceedings of the IEEE/CVF Conference on Computer Vision and Pattern Recognition, pp. 1546-1555. 2024.

[3] Pinyoanuntapong, Ekkasit, Muhammad Usama Saleem, Pu Wang, Minwoo Lee, Srijan Das, and Chen Chen. "BAMM: bidirectional autoregressive motion model." In European Conference on Computer Vision, pp. 172-190. Cham: Springer Nature Switzerland, 2024.

[4] Jiang, Biao, Xin Chen, Wen Liu, Jingyi Yu, Gang Yu, and Tao Chen. "Motiongpt: Human motion as a foreign language." Advances in Neural Information Processing Systems 36 (2023): 20067-20079.

[5] Zhang, Yaqi, Di Huang, Bin Liu, Shixiang Tang, Yan Lu, Lu Chen, Lei Bai, Qi Chu, Nenghai Yu, and Wanli Ouyang. "Motiongpt: Finetuned llms are general-purpose motion generators." In Proceedings of the AAAI Conference on Artificial Intelligence, vol. 38, no. 7, pp. 7368-7376. 2024.

**Ethical Concerns:**

["NO or VERY MINOR ethics concerns only"]

**Final Justification:**

The rebuttal further address some of my concerns (please refer to my comments below.)

Overall I think it's a paper that meets the minimum requirement of an accepted paper.

**Limitations:**

yes

**Paper Formatting Concerns:**

no concerns as far as I am aware

**Quality:**

3

**Strengths And Weaknesses:**

Pros:

1. The paper is well-written and clearly articulated.
The figures are highly intuitive and effectively enhance understanding; for instance, Figure 2 provides an excellent and comprehensive overview of the algorithm. The algorithm is presented in a way to be easily and readily comprehensible to readers.

2. The experiment section is very complete and fully supports the claim of the paper.

The paper holds a full collection of more than 10 baseline algorithms and collects a wide range of metrics such as beat alignment and diversity. A comprehensive user study is also included.

The ablation study is also well prepared, clearly showing the impact of each component in the model.

Cons:

1. The paper would benefit from a more detailed exposition on the intuition behind the choices made for the tokenizer design.
A comparative analysis of different tokenization strategies is needed to fully justify the proposed approach.


What happens if we just use a multi-scale tokenizer without the separate encoding for different body parts?

What happens if we just use multi-head vqvae instead of multi-scale vqvae?


2. The contribution might be seen as incremental.

While the proposed algorithm shows improved performance compared to the baselines, it might be seen as an application of existing tokenization strategy on a specific domain (motion generation). Multi-scale tokenization and prediction strategy has been proposed and as discussed in the paper [1].

And the body part based tokenization strategy is designed for the motion generation problem and does not seem to be generalizable to other domains.

[1] Tian, Keyu, Yi Jiang, Zehuan Yuan, Bingyue Peng, and Liwei Wang. "Visual autoregressive modeling: Scalable image generation via next-scale prediction." Advances in neural information processing systems 37 (2024): 84839-84865.



Based on the quality of the experiments and paper writing, I lean towards borderline acceptance.

However, given concerns regarding the novelty of the contribution, I will not actively champion this submission.

---

> ### Author Rebuttal · Authors · 2025-07-31
>
> Weakness 1:
>
> We appreciate the insightful comment. The intuition behind our tokenizer design choices stems from a trade-off between computational inefficiency—due to overlapping information across different tokenizer scales—and the loss of multi-scale information resulting from an overly sparse selection of scale levels. Inspired by feature pyramid networks, we opted for a power-of-2 scaling factor to balance these considerations effectively
>
> Q1:
>
> If we just use a multi-scale tokenizer without the separate encoding for different body parts?
> Response: Thank you for this insightful question. We actually evaluated this exact configuration in our ablation studies (Table 2, Line 1), where we observed significant performance degradation compared to using separate encodings. This performance drop likely stems from interference patterns between different body parts' motions. Our findings align with observations in prior work like EMAGE, which similarly demonstrates the importance of separate encodings for different body regions.
>
> Q2:
>
> What happens if we just use multi-head VQ-VAE instead of multi-scale VQ-VAE?
> Response: Thank you for raising this important comparison. The multi-head VQVAE (as implemented in MSMC-TTS) can indeed capture richer single-scale patterns through multiple codebooks. However, the multi-head VQ-VAE is still not able to capture and fuse multi-scale motion patterns since the multi-head VQ-VAE does not contain the process of extracting motion patterns with different scales in the latent representation level, let alone fusing multi-scale features of motion patterns. The fundamental architectural difference means that multi-head VQVAE is not suitable in both capturing hierarchical motion patterns and integrating them effectively than multi-scale VQVAE.
>
> Weakness 2:
>
> Thank you for your comment. We apologize for not clearly articulating our contributions to both the motion generation field and the broader research community. As mentioned in Line 35, the core idea of pyramidal tokenization in PyraMotion is inspired by the concept of pyramidal multiscale design in computer vision, drawing not only from [1] but also from feature pyramid networks [2].
> We would like to clarify that the key novelties of this paper are:
> 1. Identifying the research gap in single-scale tokenization for temporal data generation.
> 2. Proposing pyramidal-tokenization-based models, such as APVQVAE and PyraMotion, to address this limitation.
> Furthermore, we argue that the single-scale tokenization gap is not unique to motion generation but also exists in other 1D temporal sequence generation tasks, such as trajectory generation and stock price prediction. Thus, we believe our proposed Attentional Pyramidal VQ-VAE has the potential for broader applicability in capturing and fusing multiscale temporal patterns across different domains.
>
> We will incorporate an analysis of the broader implications of this work in the introduction section to better highlight its impact.
>
> [1] Tian, Keyu, Yi Jiang, Zehuan Yuan, Bingyue Peng, and Liwei Wang. "Visual autoregressive modeling: Scalable image generation via next-scale prediction." Advances in neural information processing systems 37 (2024): 84839-84865.
>
> [2] Tsung-Yi Lin, Piotr Dollár, Ross Girshick, Kaiming He, Bharath Hariharan, and Serge Belongie. Feature pyramid networks for object detection. In Proceedings of the IEEE conference on computer vision and pattern recognition, pages 2117–2125, 2017.
>
> Question 1:
> Thanks for your question. The inference time of the single-scale model EMAGE and the multi-scale model Pyramotion on a motion of more than 5 min is listed below. The real-time ratio of both our model and EMAGE is far less than 1; thus, it is streamable.
>
> |       |  Inference Time |
> |-------|-------|
> | EMAGE       |  22 s |
> | PyraMotion  |  41 s |
>
> Question 2:
>
> Thanks for pointing out our imprecise statement. We will correct our statement to “Existing systems tokenize gestures within a single scale manner.”
>
> For the separate tokenization issue. Thanks for sharing your precious experience about it. As I mentioned above, in our ablation experiment, we first evaluated the performance of tokenization on a unified body and recorded it in Table 2, Line 1. The experiment results show that such separation largely contributes to the final performance. Such a conclusion is also validated in the paper EMAGE. As a result, such differences in the separation strategy may be related to model types and data distributions. The detailed research about it is valuable and will contribute to the research community. We will include it in our future work analysis.

---

> ### Author Response · Authors · 2025-08-07
>
> I hope this message finds you well. I appreciate your thoughtful comments and have carefully responded to each of your concerns in my response. I hope my responses meet your expectations. Please don’t hesitate to let me know if any further clarifications or adjustments are needed.

---

> > ### Comment · Reviewer_CqFS · 2025-08-08
> > **thank you for the rebuttal**
> >
> > I appreciate the authors' rebuttal and this addresses some of my concerns.
> >
> > Overall I think motion generation with the use of tokenizer is a very exciting and interesting work direction.
> > And lot of new papers are showing up. This paper presents a novel method which further builds on top of this line of research.
> >
> >
> > I still think some of the ablation study will still be very helpful (multi-head / multi-scale tokenzier), and overall I tend to recommend for acceptance. (And I hope more discussions and ablation will be added to the final draft.)

---

> > > ### Author Response · Authors · 2025-08-09
> > >
> > > Thank you for your insightful comments regarding our work's contribution to the motion generation community. We agree that the application of tokenizers in motion generation is highly meaningful and holds considerable potential for further exploration.
> > >
> > > Regarding the exploration of multi-head and multi-scale tokenizers, we believe these are valuable suggestions that could significantly advance tokenization-based motion generation. We will surely incorporate more extensive discussion and ablation studies on these points in our final version once it is accepted.

---

### Official Review · Reviewer_JaLi · 2025-06-28

**Clarity:** 3
**Significance:** 3
**Originality:** 4
**Rating:** 4
**Confidence:** 3

**Summary:**

This paper proposes a novel multi-scale pyramidal framework named PyraMotion to achieve audio-driven full-body human gestures generation. It introduces the Attentional Pyramidal VQ-VAE (APVQ-VAE), which encodes gestures at multiple temporal scales using pyramidal temporal convolution networks and reconstructs motion via attention-based fusion of multi-scale token sequences. On top of this, PyraMotion introduces a pyramidal token predictor that uses audio, text, speaker identity, and partial motion hints to predict multi-scale gesture tokens for different body parts, enabling fine-grained and coherent gesture synthesis. The experiments demonstrate the effectiveness and superior performance of the proposed framework.

**Questions:**

(1) Given that the proposed framework introduces additional architectural components (e.g., multi-scale TCNs), it is necessary to compare its computational complexity and runtime with previous works. Regarding the optimal number of pyramid layers, Table 4 suggests that four layers yield the best performance. However, I noticed inconsistencies between the results in Table 4 (4 Layers) and Table 3 (APVQ-VAE). Specifically, the Upper metric should be 2.612 ± 0.023, and the BC and LVD values should be ±0.015 and ±0.019, respectively. The authors are expected to clarify whether these are typographical errors or explain the discrepancy.

(2) Regarding the content of the paper, the “Limitation, Future Work” section lists only one limitation, and the prefix “1)” is unnecessary. In Line 8–9, the claim that “such non-verbal signals provide more information than voice and context” may be overly absolute. The authors should either clarify the specific scenarios where this applies or rephrase the statement more cautiously. In Line 293–294, the authors should provide further details on the 18 participants in the perceptual study, such as gender, age, or research background, as these factors could influence subjective evaluation outcomes.

(3) The font size in Figure 2 is too small, which significantly impacts readability and visual clarity. It is recommended to enlarge the figure to improve legibility. Additionally, whether $L_{vel}$ and $L_{acc}$ in Equation 6 are commonly used loss components in motion generation tasks requires further explanation from the authors.

**Ethical Concerns:**

["NO or VERY MINOR ethics concerns only"]

**Final Justification:**

The authors have addressed my concerns, and I think this solid work meets the acceptance requirements. Therefore, I keep my original score as 4.

**Limitations:**

yes

**Paper Formatting Concerns:**

The paper does not have any major formatting issues.

**Quality:**

4

**Strengths And Weaknesses:**

Strengths:
This paper presents a high-quality and well-executed framework for audio-driven full-body gesture generation. Its main contribution lies in the introduction of the Attentional Pyramidal VQ-VAE, a novel architecture that encodes motion patterns across multiple temporal scales, and its integration into the PyraMotion framework for multi-modal gesture synthesis. The paper is clearly written, well-structured, and supported by thorough experiments. The proposed method achieves state-of-the-art results across various benchmarks, demonstrating both originality and effectiveness.

Weakness:
This model introduces architectural complexity, so it is necessary to provide PyraMotion with a comparison of computational complexity and computational time to other methods. Certain sections of the text and figures/tables necessitate additional adjustments, which will be elaborated below.

---

> ### Author Rebuttal · Authors · 2025-07-31
>
> Weakness 1:
>
> Thank you for pointing out the complexity of our method. We will add an analysis comparing the time complexity of our work with other single-scale methods to the appendix in the final version.
>
> Theoretically, the computational complexity of APVQ-VAE is:
> $$O\left(\sum_{p=1}^{P}(2^pd)\left(\frac{Nd}{2^p}\right)\right) = O(PNd^2)$$
> where $d$ is the latent dimension, $N$ is the total number of frames, and $P$ is the number of pyramid layers. Thus, the time complexity of APVQ-VAE is a constant multiple of vanilla VQ-VAE.
>
> For the PyraMotion model, after the pooling operation, the time complexity at each step follows a geometric series. For example, in the TCA operation (Equation 12), the single-scale time complexity is:
> $$O(Nd^2 + N^2d)$$
> where $N$ is the clip length and $d$ is the hidden dimension. The multi-scale Temporal Cross-Attention operation has complexity:
> $$O\left(\sum_{p=0}^{P}\left(\frac{Nd^2}{2^p} + \left(\frac{N}{2^p}\right)^2d\right)\right) \sim O(2Nd^2 + \frac{4N^2d}{3}) < 2O(Nd^2 + N^2d)$$
> This shows PyraMotion's time complexity is less than 2× that of its single-scale version.
>
> Specifically, we measured the inference times for single-scale EMAGE and multi-scale PyraMotion on over 5 minutes of motion data. The experimental results validate these theoretical conclusions.
>
> |       |  Inference Time |
> |-------|-------|
> | EMAGE       |  22 s |
> | PyraMotion  |  41 s |
>
> Question 1:
>
> Thank you for pointing out the statistical typo. For time-complexity analysis, please refer to our response to Weakness 1. Regarding the discrepancy between Table 4 and Table 3, we sincerely apologize for this oversight. After thorough verification of our local statistics, we confirm that Table 3 contains the correct values. We will immediately correct Table 4 accordingly.
>
> Question 2:
>
> Thank you for your insightful suggestions.
>
> Regarding limitations: Due to page constraints, we initially provided only a brief discussion of limitations. We will add another key limitation in the final version: Our current model underperforms existing emotion-disentangled approaches in generating emotional expressions from vocal affect, as it relies heavily on the training data's emotion distribution rather than explicitly extracting emotion representations from audio.
>
> Concerning the claim about non-verbal signals, we appreciate your identifying the imprecise phrasing. We will modify the statement to: "Such non-verbal signals provide distinct information that neither voice nor context can adequately express."
>
> Regarding participant demographics: Thank you for the suggestion. Our participant pool consisted of:
>
> Age: 12 participants (18-24 years), 6 participants (25-34 years)
>
> Gender: 8 male, 10 female
>
> Education: 5 Bachelor's, 5 MPhil students, 8 PhD students
>
> Question 3:
>
> Thank you for your constructive comments.
>
> For Figure 2, we will revise the figure to improve the clarity and readability of the overall structure.
> Regarding the loss terms in Equation 6:
> $L_{vel}$ appears in DiffSHEG, EMAGE, Habibie et al., etc.
> $L_{acc}$ appears in EMAGE, Habibie et al., CaMN, etc.
> We will ensure all these references are properly included in the final version.

---

> > ### Comment · Reviewer_JaLi · 2025-08-06
> >
> > Dear Authors:
> >
> > Thank you for the detailed rebuttal. The authors have provided a thorough analysis of the computational complexity and runtime. My concerns have been well addressed, and the authors' response is satisfactory.
> >
> > Therefore, I am inclined to accept this paper and will maintain my original score.
> >
> > Best regards

---

### Official Review · Reviewer_VSf9 · 2025-07-01

**Clarity:** 3
**Significance:** 3
**Originality:** 4
**Rating:** 5
**Confidence:** 4

**Summary:**

This paper proposes PyraMotion, a framework for co-speech gesture generation that integrates multi-scale motion tokenization with attention-based fusion. It introduces an adaptive pyramid-structured VQ-VAE model, called APVQ-VAE, which captures motion patterns across different temporal scales and body parts. Extensive experiments demonstrate superior performance over state-of-the-art methods in both objective metrics and subjective perception.

**Questions:**

For a motion sequence x with shape of (L,D), where L is the time length and D is the feature dim. What is the output if feed x into the proposed APVQ-VAE?

**Ethical Concerns:**

["NO or VERY MINOR ethics concerns only"]

**Final Justification:**

I appreciate the authors' detailed rebuttal, which has addressed my main concerns. The work is technically thorough, with high reproducibility. It demonstrates significant innovation and has great potential. I believe that with the open-source release, it will contribute to advancing co-speech gesture generation. Therefore, I raise my rating to Accept.

**Limitations:**

As weakness.

**Paper Formatting Concerns:**

No Formatting Concerns.

**Quality:**

3

**Strengths And Weaknesses:**

Strengths:

1. The APVQ-VAE enables multi-scale modeling of motion patterns, and the attention maps provide interpretable insights into how different body parts attend to different temporal resolutions.

2. The paper includes thorough ablation studies, comparisons with SOTA methods, and subjective human evaluations, all supporting the effectiveness of the proposed approach.

3. The authors provide sufficient visualizations through videos and conduct comprehensive quantitative experiments.

Weakness:

1. Line 28 is imprecise. Typically, VQ-VAE can encode T frames of motion into T/d tokens, where d is the downsampling rate. Therefore, existing VQ-VAE methods do not model only single-frame motion. It is recommended that the authors revise this statement in their rebuttal.

2. The comparison with state-of-the-art (SOTA) methods for co-speech gesture generation, such as MambaTalk and SynTalker, is missing.

---

> ### Author Rebuttal · Authors · 2025-07-31
>
> Weakness 1:
>
> Thank you for pointing out the imprecise statement. We will revise this statement to: "Existing VQ-VAE methods can only model motion in a single scale, which may not be able to model the complexity of natural human motions with varying durations."
>
> Weakness 2:
>
> Thank you for your suggestion about the comparison with the listed works. We will add the comparison with these works to the final version of our paper. Moreover, we would like to clarify that, compared with these works, the claim that our method achieves and outperforms state-of-the-art holistic generation work is still valid.
>
> |       |  FGD  |   BA  |  Div  |  MSE  |  LVD  |
> |-------|-------|-------|-------|-------|-------|
> | Mamba-Talk     |  5.366 | 7.812 | 13.05  | 6.289 | 6.897 |
> | SynTalk        |  6.413 | 7.971 | 12.721 |   -   |   -   |
> | SynTalk(noboth)|  4.687 | 7.363 | 12.425 |   -   |   -   |
> | Ours           |  4.612 | 7.420 | 13.420 | 7.176 | 7.270 |
>
> Question 1:
>
> Thanks for raising this question. if we feed a motion sequence $x_{in} \in \mathbb{R}^{L\times D}$ into APVQ-VAE, the output will also be a reconstructed motion sequence $x_{out} \in \mathbb{R}^{L\times D}$. Such an input-output format is consistent with traditional VQVAE. However, the difference between APVQ-VAE and VQ-VAE is that, VQ-VAE reconstructed the $L\times D$ motion sequence by encoding it with encoder network into a token series $T \in \mathbb{R}^{L_T\times 1}$, and learns a codebook $C \in \mathbb{R}^{|C|\times d}$ and reconstruct it by decoding the token series with a decoder network. However, our APVQ-VAE encode the $L\times D$ motion sequence into a series of pyramidal token series, the token series in $p^{th}$ layer is $T_p \in \mathbb{R}^{(L_T/2^p)\times 1}$. The decoder network will decode these token series and generate $p$ motion sequences, which will be attentively fused into a final motion sequence representation $x_{out} \in \mathbb{R}^{L\times D}$

---

> ### Comment · Reviewer_VSf9 · 2025-08-06
>
> Thank you for your response, which addressed my main concerns. I have a few follow-up questions regarding some details, and I would appreciate further clarification if possible.
>
> Regarding the full-body VQ-VAE modeling, how do you maintain coordination between different body parts while also preserving the fine-grained motion details for each body part?
>
> In Line 122, it is mentioned that the upper body, lower body, face, and hands are encoded into different latent spaces. Does this mean that the four APVQ-VAE models encode these body parts separately? If so, are the encoding of different body parts completely independent?
>
> If they are independent, could the diagram in Figure 2 be revised in future versions to make this clearer?
> Is the relationship between different body parts modeled by the subsequent PyraMotion sequence model? How effective is this approach, and are there any potential issues?
>
> If the encoding of body parts is not completely independent, how does the APVQ-VAE handle the relationships between different body parts during encoding?
>
> The proposed APVQ-VAE shows great potential. It would be a significant contribution to the community if it were open-sourced. Would the authors consider making it publicly available?

---

> > ### Author Response · Authors · 2025-08-06
> >
> > Thanks for your thoughtful and insightful questions. It would be a pleasure for us to give further clarification on your reviews. I will mark your questions as question 1,2,3,4,5 separately and respond to them in total.
> >
> > Response:
> >
> > 1. In response to Question 2, we would like to clarify that the encoding of different body parts is completely independent. Since the holistic body motion data can be seen as a 206-dimensional vector, where facial expression is represented by a 100-dimensional vector, and full-body motion is represented by a 106-dimensional vector, denoting the parameters of joint position and rotations. The four APVQ-VAEs are trained on different parts of the 206-dimensional body representation without overlapping, and thus are fully independent.
> >
> > 2. In response to Question 3, yes, for sure, we will review the Figure 2 diagram for a clearer illustration of the overall workflow. And yes, the correlation information among different body parts is mainly modeled by the subsequent PyraMotion model. To analyze the effectiveness of our PyraMotion, we would like to first elaborate on the detailed training process of it.
> >
> > After obtaining four APVQ-VAEs for different body parts, for the speech-motion data pairs in the dataset, we first encode the motion into four token series using the encoders from these APVQ-VAEs. Since APVQ-VAE works on encoding motion into a token series and decoding the token series into a reconstructed motion sequence, we would like to clarify that the correlation information among different body parts is compressed into the four multi-scale token series.
> >
> > Then, we utilize PyraMotion to model the audio features and generate four token series with the target of the encoded token series above. And we construct the loss function from the differences between the audio-based generated token series and the encoded token series for updating the parameters of PyraMotion.
> >
> > That is to say, we believe the effectiveness of this workflow in modeling cross-body parts correlations depends on two main factors: the capability of PyraMotion in generating different body parts token series with modeling cross-body parts correlations, and the capability of APVQ-VAEs in compressing motion sequences with as little information loss as possible.
> >
> > In PyraMotion, we design a feature integrating mechanism in Equation (12), where we generate multi-scale body-part features by integrating the full-body features, which enables the workflow to generate correlated body part token series. The performance comparison with the w/o Full-Body Latent variant in Table 2 ablation experiments illustrates the effectiveness of this structure.
> >
> > While for the APVQ-VAE performance, considering the better reconstruction performance of APVQ-VAEs compared with vanilla VQ-VAEs shown in Table 3 in the paper, the APVQ-VAE explicitly outperforms existing methods in tokenizing motion sequences with less information loss.
> >
> > In conclusion, we would like to clarify that the proposed APVQ-VAE and PyraMotion are effective in capturing correlation information between different body parts compared with vanilla VQ-VAE. However, there is still some space for improvement. One observation based on the existing work, EMAGE, and our work is that overly focusing on integrating the correlation among different body parts could potentially cause a decrease due to the interference of different body parts' motion distributions. While fully independent modeling of each body part would cause unnatural behavior. Thus, a self-adaptive mechanism of adjusting the ratio between these two perspectives would be potentially beneficial for future studies. Such a discussion will be added to the future work.
> >
> > 3. For Question 1, after analyzing the above questions, we could reach a conclusion about how to balance the correlation between different body parts and how to preserve fine-grained body motions. The APVQ-VAE is applied to different body parts independently. For a single body part, the multi-scale token series encoded by APVQ-VAE preserves fine-grained body motions into a multi-scale token series. While for correlation information, the generated four multi-scale token series preserve the full-body motion information and guide the PyraMotion structure to generate correlated token series through an explicitly defined correlated attentive feature fusion mechanism defined as Equation (12).
> >
> > 4. For Question 5, yes, for sure we will open source both our source code and pretrained model after acceptance. And the source code is already attached in the supplementary files.

---

> > > ### Comment · Reviewer_VSf9 · 2025-08-07
> > >
> > > I appreciate the authors' response and code.
> > > That addresss my main concerns.

---

> > > > ### Author Response · Authors · 2025-08-07
> > > >
> > > > Thank you for your approbation. I appreciate such a discussion that helps clarify the content and supports the knowledge transmission. I would like to express my greatest gratitude for your recognition of our contribution to the community. This is a huge effort for me and my mate to come this far, and I would appreciate it if our effort pays off. Thank you again for your acknowledgment.

---

### Official Review · Reviewer_QWGa · 2025-07-03

**Clarity:** 3
**Significance:** 3
**Originality:** 2
**Rating:** 4
**Confidence:** 4

**Summary:**

This paper introduces PyraMotion, a new framework for generating natural and expressive full-body co-speech gestures from audio inputs. Unlike existing methods that predict frame-wise motion tokens, PyraMotion captures multi-scale motion patterns across different body parts by leveraging a pyramidal structure. Central to this approach is the proposed Attentional Pyramidal VQ-VAE (APVQ-VAE), which encodes gesture sequences at varying temporal scales into a shared discrete latent space and reconstructs gestures using an attention-based fusion mechanism. The system also includes a pyramidal token predictor that integrates audio and textual features to accurately predict motion tokens for various body regions. Objective and subjective evaluations demonstrate that PyraMotion significantly outperforms baseline methods in gesture realism, diversity, and synchronization. Ablation studies confirm the importance of its key components, particularly the attention mechanism and multi-scale design.

**Questions:**

1. The limitation section is relatively brief and does not show concrete examples of where the model fails or underperforms. For instance, are there specific types of speech (e.g., monotone, emotional) that the model struggles with?
2. The paper compares mainly with prior VQ-VAE-based systems. However, recent approaches such as DiffSHEG or transformer-based generators may offer different trade-offs. Can the authors comment on these directions, and whether PyraMotion could be combined with them?
3. Why were recent strong baselines such as FreeMotion, Emotional Speech-driven Body Animation, and DiffSHEG not included in the comparison? Please clarify the rationale for not including these baselines. If they were omitted due to incompatible inputs, unavailable code, or other technical barriers, a discussion would help justify the scope of comparison. Ideally, adding results for at least one of them would greatly strengthen the paper.

**Ethical Concerns:**

["NO or VERY MINOR ethics concerns only"]

**Final Justification:**

Although the rebuttal has clarified my major concerns and improved the paper, the level of novelty reamins average. Therefore, I maintain my borderline accept recommendation.

**Limitations:**

Yes.

**Paper Formatting Concerns:**

No major formatting issue.

**Quality:**

2

**Strengths And Weaknesses:**

# Strengths
- The paper is generally well-written and easy to follow.
- The proposed PyraMotion framework is well-motivated, and the design of the Attentional Pyramidal VQ-VAE (APVQ-VAE) is carefully validated through quantitative and qualitative evaluations.

# Weaknesses
- While the pyramidal motion integration is new in this domain, the use of pyramidal architectures and attention mechanisms is conceptually borrowed from established techniques in computer vision.
- While the paper compares against a number of established methods, it omits several recent and relevant approaches such as FreeMotion, Emotional Speech-driven 3D Body Animation via Disentangled Latent Diffusion, and DiffSHEG. These newer methods represent the current frontier in text/audio-to-motion synthesis and may offer strong performance baselines, particularly in diffusion-based frameworks. The absence of these baselines weakens the comparative analysis and may limit the ability to assess PyraMotion's true performance advantage.

---

> ### Author Rebuttal · Authors · 2025-07-31
>
> Weakness 1:
>
> Thank you for your insightful comment. As stated in the introduction section, we drew inspiration from the feature pyramid to construct the pyramidal VQ-VAE for tokenizing 3D holistic motion in a multi-scale manner. But we would like to clarify that the novelties of this paper are: Identifying the single-scale tokenization research gap and proposing pyramidal-tokenization-based models such as APVQVAE and PyraMotion to address it.
>
>
> Weakness 2 / Question 3:
>
> Thank you for your insightful suggestion. We would like to clarify that DiffSHEG is included as a comparison method in Line 10 of Table 1.
>
> Regarding the other two methods, there are specific reasons for not including them in our comparisons. For FreeMotion, its primary task focuses on text-to-motion generation, which produces motion from descriptive text. In contrast, our task is co-speech motion generation, where motion is generated from audio input. The transcripts used in our work represent the semantic content of the audio rather than motion descriptions. Therefore, FreeMotion may not be suitable as a direct comparison method, though we will include it in our related works section.
>
> As for AMUSE, in the initial stages of this paper, we primarily selected state-of-the-art methods that incorporate VQ-VAE and its variant structures. We appreciate your suggestion and will add AMUSE to our comparison methods in the final version to ensure a more comprehensive evaluation.
>
> Question 1:
>
> Thank you for your insightful suggestion. Due to page limit considerations, we only provided a brief introduction to the paper's limitations. Here we would like to introduce one additional limitation for the final version: Our current model still underperforms existing emotion-disentangled models in generating emotional expressions based on vocal emotion. It relies heavily on the emotion distribution of the training data rather than explicitly extracting emotion representations from audio.
>
> Question 2:
>
> Thank you for your valuable question. The core idea of PyraMotion involves tokenizing motion representations at multiple scales and adaptively combining these multi-scale features to reconstruct motion.
>
> In contrast, diffusion-based or transformer-based methods do not include this tokenization step. These methods represent motion patterns through hidden state embeddings in latent space, which are inherently frame-wise representations.
>
> To adapt PyraMotion's approach to diffusion-based or transformer-based methods, we could first fuse motion blocks of different lengths to create multiple motion representation sequences with varying granularities. We could then use the diffusion or transformer model to generate these multi-scale motion representation sequences before integrating them through attention mechanisms.

---

> > ### Comment · Reviewer_QWGa · 2025-08-09
> >
> > Thank you for your detailed rebuttal. I think the rebuttal has addressed my main concerns. I hope that the authors would add AMUSE for comparison and include more discussion on the limitation of their method in the final version of their paper. For now, I would keep my score.

---

> > > ### Author Response · Authors · 2025-08-09
> > >
> > > Thank you for your positive response. For the comparison with AMUSE and discussion on the limitation, we will surely add them in the final version for a more comprehensive comparison and discussion, as stated in the rebuttal once it is accepted.

---

> ### Author Response · Authors · 2025-08-07
>
> I hope this message finds you well. I appreciate your thoughtful comments and have carefully responded to each of your concerns in my response. I hope my responses meet your expectations. Please don’t hesitate to let me know if any further clarifications or adjustments are needed.

---

### Official Review · Reviewer_ZG2m · 2025-07-07

**Clarity:** 1
**Significance:** 2
**Originality:** 2
**Rating:** 3
**Confidence:** 4

**Summary:**

The paper presents PyraMotion, a speech-driven gesture generation method that leverages an attentional pyramid-structured motion integration mechanism. This approach captures expressive gestures by modeling motion patterns of varying durations across different frame lengths and body parts, enabling the system to generate gestures that are temporally and spatially consistent with speech.

**Questions:**

The primary concerns are outlined in the Weaknesses section above.

- Regarding the actor Scott from the BEAT-2 dataset: was this a held-out actor during training? If not, could you please include qualitative examples from an actual held-out actor to better assess generalization?

**Ethical Concerns:**

["NO or VERY MINOR ethics concerns only"]

**Final Justification:**

While the rebuttal addressed some concerns, the work still lacks comprehensive SOTA comparisons, limiting the strength of its claims.

**Limitations:**

Yes

**Paper Formatting Concerns:**

No concerns.

**Quality:**

1

**Strengths And Weaknesses:**

Strengths:
- The paper proposes a novel technique that encodes motion patterns at varying temporal scales into a shared discrete latent space as motion pattern tokens, enabling multi-scale gesture representation.

- The authors introduce a novel attentional pyramidal VQ-VAE architecture to tokenize motion across different temporal resolutions, allowing the model to capture body movements over time.

Weaknesses:
- The generated gestures often appear jittery, with broken lip-sync, and unnatural finger poses.

- Figures:
-- Figure 2 is difficult to interpret due to overlapping arrows and dense layout.
-- Figure 3 uses inconsistent font sizes, with text hard to read.
-- The attention map lacks a color legend, reducing interpretability of visualized patterns.

- The chosen metrics: FGD, Diversity, Beat Alignment (BA), and MSE, are standard for speech-to-gesture tasks, but may not effectively capture the multi-scale motion modeling claims made in the paper. The use of task-specific or scale-sensitive metrics would support the evaluation.

- The evaluation omits several recent SOTA methods, such as: HOLOGEST (Cheng et al., 2025), AMUSE (CVPR 2024). ConvoFusion (CVPR 2024), RAG-Gesture (CVPR 2025). These should be included for a comprehensive comparison.

- The user study relies on older methods like CaMN, despite the availability of newer, more relevant baselines. While EMAGE (CVPR 2024) is included, more recent methods should be part of the perceptual comparison to validate the model's effectiveness.

- According to Appendix A, the model requires 3 days of training, which is high compared to recent SOTA methods. This raises concerns about computational efficiency, reproducibility, and accessibility.

---

> ### Author Rebuttal · Authors · 2025-07-31
>
> Thanks for your thorough and insightful comments. We will respond to them one by one as follows:
>
> Weakness 1:
>
> Thank you for pointing out the unsatisfying performance of our method in terms of jittering, mouth movements, and finger movements. We acknowledge that there is a gap between the current performance and a satisfactory outcome. However, we would like to clarify that both the conducted user perceptual and statistical experiments have validated that our generated motion outperforms existing classical and state-of-the-art methods.
>
> Weakness 2:
>
> Thank you for pointing out the flaws in our used figures. We will update them with clearer figures in the final version.
>
> Weakness 3:
>
> Thank you for raising your concerns about the used metrics. We adapt these metrics following several state-of-the-art works, such as DiffSHEG, EMAGE, and ProbTalk. We agree with you that a metric measuring multi-scale motion is valuable. However, we would like to clarify that our current metrics, such as FGD, Beat Align, and MSE, are sufficient to evaluate the overall similarity, smoothness, and accuracy of the generated motion and reach the claims made in our paper.
>
> Weakness 4:
>
> Thank you for listing these SOTA methods. On one hand, we would like to clarify that we adopt EMAGE and ProbTalk as our SOTA method mainly because the tasks of these two methods are holistic body motion generation, including both full body and face, while the given works are mainly full body generation without face motion generation. On the other hand, we would like to argue that both EMAGE and ProbTalk are published in the same period as most of the listed works, as state-of-the-art. Moreover, RAG-Gesture was published in June 2025, when our work had already been submitted to NeurIPS, and its code is not yet available. Thus, RAG-Gesture may not be suitable to be counted as a SOTA work before our paper.
>
> To accomplish a more thorough comparison with the listed state-of-the-art methods, we will add comparisons with these methods in the final version of the paper.
>
> Due to the limited duration of the rebuttal, we have only recorded two of the mentioned methods. The experiment results show that our method still outperforms these state-of-the-art methods, even RAG-Gesture in some circumstances, which further validates the performance of our method.
>
> |       |  FGD  |   BA  |  Div  |  MSE  |  LVD  |
> |-------|-------|-------|-------|-------|-------|
> | HOLOGEST    | 5.347 | 7.957 | 14.15 |   -   |   -   |
> | RAG-Gesture | 8.08  | 7.34  | 11.97 |   -   |   -   |
> | Ours        | 4.612 | 7.420 | 13.42 | 7.176 | 7.270 |
>
>
> Weakness 5:
>
> Thank you for your suggestions. We agree that comparing with more state-of-the-art work would strengthen the validation of our paper's effectiveness. We will add the comparison with ProbTalk in the final version.
>
> However, we would like to clarify that the current experiments also validate our claims. CaMN is a classic and well-performing method that has been used in user perceptual experiments in several recent works, such as EMAGE and MambaTalk. As a result, we included CaMN in our user perceptual comparison. The user perceptual results in our paper also validate that CaMN is necessary, as it outperforms EMAGE in some cases.
>
> Meanwhile, we selected EMAGE, a state-of-the-art paper published in CVPR 2024, as our recent comparison primarily because it is a VQVAE-based state-of-the-art method, which helps illustrate the difference between our proposed method and traditional VQVAE-based approaches. Since ProbTalk is also a VQVAE-based method, we will add it to the user perceptual comparison experiments.
>
> Weakness 6:
>
> Thank you for your concerns about the training time. We argue that the 3-day training time does not affect the paper’s quality in terms of reproducibility and accessibility. We will release both the source code and pre-trained parameters on GitHub.
> As for computational efficiency, the additional granularity layers cause an increase, which we discuss from Lines 281 to 287 on Page 8.
>
> Theoretically, the computational complexity of APVQ-VAE is $O(\Sigma_{p = 1}^{P}(2^pd)(Nd/2^p)) = O(PNd^2)$, where $d$ is the latent dimension, $N$ is the total number of data frame, and $P$ is the number of pyramid layers. Thus, the time complexity of APVQ-VAE is a constant times that of vanilla VQ-VAE.
>
> For the PyraMotion model, after the pooling operation. The time complexity of each step satisfies the form of a geometric series. For example, in the TCA operation in Equation (12), the time complexity of single scale is $O(Nd^2 + N^2d)$, where $N$ is the length of the total clip, while $d$ is the hidden dimension. For multi-sclae Temporal cross-attention operation, the time complexity is $O(\Sigma_{p=0}^{P}(Nd^2/2^p + (N/2^p)^2d)) \sim O(2Nd^2+4N^2d/3) < 2O(Nd^2 + N^2d)$. Thus, we can conclude that the time complexity of PyraMotion is less than 2 times that of the single-scale version of Pyramotion.
>
> Technically, the inference time of the single-scale model EMAGE and the multi-scale model PyraMotion on more than 5 min motion is listed below. The experimental result validates the above theoretical results.
>
> |       |  Inference Time |
> |-------|-------|
> | EMAGE       |  22 s |
> | PyraMotion  |  41 s |
>
> We will add the above analysis to our appendix in the final version.
>
> Question 1:
>
> Thank you for your suggestion. To ensure a fair comparison with existing methods, we follow the training/test/validation split strategy used in prior works like EMAGE, ProbTalk, and CaMN, which do not include a held-out actor. We will add clarification regarding held-out actors in the final version.

---

> ### Author Response · Authors · 2025-08-07
>
> I hope this message finds you well. I appreciate your thoughtful comments and have carefully responded to each of your concerns in my response. I hope my responses meet your expectations. Please don’t hesitate to let me know if any further clarifications or adjustments are needed.

---

> > ### Comment · Reviewer_ZG2m · 2025-08-09
> > **Response to rebuttal**
> >
> > I thank the authors for a detailed rebuttal and for addressing my concerns about improving the figures, reporting the training footprint, and clarifying the held-out actor process to ensure comparability with prior work.
> >
> > However, I remain concerned about the SOTA comparisons. While RAG Gesture is a concurrent work and need not be included, Hologest (3DV 2025), AMUSE, and ConvoFusion (both CVPR 2024) are important baselines for a faithful validation. CAMN, being a precursor to EMAGE, is not an appropriate baseline. We also observe that Hologest outperforms Pyramotion on beat alignment and diversity. It is unclear why comparisons are limited to VQ-VAE based methods, the goal is high-quality motion regardless of architecture. In addition, the rebuttal does not quantitatively validate the multiscale modeling, standard metrics do not isolate the gains resulting from the multiscale design. Overall, the work requires evaluation against current SOTA methods to substantiate the claims and verify relative performance (Hologest/ ConvoFusion/ AMUSE, etc.) in both quantitative and perceptual studies. I raise my score to 3.

---

> > > ### Author Response · Authors · 2025-08-09
> > >
> > > Thanks for your suggestions and for raising the score. We sincerely appreciate it as an acknowledgment of our work. In terms of your concerns about the SOTA comparison, we will add Hologest, AMUSE, and ConvoFusion in the final version for further validation, as we stated in the rebuttal.
> > >
> > > For CaMN, we select it as the classical baseline method instead of the SOTA baseline method for a thorough comparison with the holistic motion generation community.
> > >
> > > For comparison with VQVAE-based methods, thanks for your advice, we will add the listed SOTA methods above for a more thorough comparison.
> > >
> > > For validating the multiscale modeling, thanks for raising this question, and we apologize for not clearly stating in the previous rebuttal comments. In the ablation experiment, Table 3, we compare multiscale VQ-VAE with vanilla VQ-VAE in learning and reconstructing the motion patterns. Since the only difference between them is the multi-scale design, we would like to clarify that the outperformance of the multiscale VQVAE in both reconstruction and generation validates the gains resulting from the multi-scale design. Such gains can be further interpreted by Figure 3, which illustrates the specific contribution of different scales towards different body parts.

---

### Note · Authors · 2025-08-16

We appreciate the time and effort of all reviewers and ACs in reviewing our work. We are grateful for the insightful and valuable comments provided, which indeed contribute to making this work more solid and impactful to the research community. We have carefully addressed all the raised concerns, and the reviewers agree that their main concerns are well addressed. The consecutive concerns about the selection of the comparison methods and the validity of the multi-scale modeling have also been responded in the following rebuttal. We think the ablation experiments can validate the effectiveness of the multi-scale modeling, and more comparisons listed in the rebuttal will be added finally.

We would like to further elaborate on how this work contributes to the community, especially in the motion generation field. Before this work, tokenization-based motion generation methods generally utilized single-scale tokens to represent the motion pattern lexicon, such as TalkSHOW. Subsequent works, such as EMAGE and ProbTalk, adopted such assumptions to represent motion lexicons in fixed-scale tokens. However, since motion patterns exist in a continuous representation space with varying temporal lengths for a complete lexicon, in this paper, we propose utilizing a multi-scale tokenization design to represent motion pattern lexicons, which can model motion patterns with different temporal lengths. Our ablation experiments on the reconstruction performance of our proposed APVQ-VAE clearly show that the multi-scale design and attentional fusion mechanism of the tokenization process significantly outperform the single-scale VQ-VAE.

To the best of our knowledge, we are the first to propose the utilization of a pyramidal-scale design for 3d motion tokenization and validate its effectiveness compared to a single-scale design. We believe this work explores a new possible direction for tokenization-based motion generation, which has the potential to diminish the existing inconsistency and unsmoothness problems present in single-scale tokenization methods.

We are confident that our paper is contributive and meaningful to the related research field, as also acknowledged by several reviewers. And we believe our work’s novel contributions and strong validation make it fit for NeurIPS, one of the most advanced conferences for AI in the world. We sincerely hope for the opportunity to get our work accepted and present it at the conference.

Thank you for your time and consideration.

---

### Decision · Program_Chairs · 2025-09-17

**Decision:**

Accept (poster)

**Comment:**

The paper introduces a method for co-speech gesture generation that incorporates a multi-scale design for 3D motion tokenization and attention-based fusion. The reviewers highlighted the following strengths: (a) a novel architecture that encodes motion patterns at varying temporal scales into a shared discrete latent space, and (b) extensive validation through ablation studies, comparisons with multiple baselines, and a user study, all supporting the effectiveness of the proposed approach.

Identified weaknesses included: missing comparisons with some SOTA methods (e.g., MambaTalk, SynTalker, Hologest, AMUSE, ConvoFusion), lack of computational complexity analysis relative to other methods, and limited discussion of alternative tokenization strategies. In the rebuttal, the authors addressed these concerns by providing computational complexity analysis, comparisons of different tokenization strategies, and additional experiments with comparisons to Hologest, MambaTalk, SynTalk, and a concurrent work (RAG-Gesture). The rebuttal clarifications satisfied four out of five reviewers. In the end, the paper received three borderline accepts, one accept, and one borderline reject (ZG2m). The borderline reviewer remained concerned about some missing comparisons (e.g., ConvoFusion, AMUSE), while the rest of the reviewers felt that the novelty of the work outweighed this weakness.

The AC agrees that further comparisons would strengthen the paper and strongly encourages the authors to include them in the final version. At the same time, the AC concurs with the majority of reviewers that the novelty of the method should be valued over incremental SOTA improvements whose absence does not diminish the scientific contribution of this work.

Thus, the AC recommends acceptance and urges the authors to carefully incorporate all reviewer feedback from both the reviews and the author-reviewer discussion when preparing the final version of the paper!